

# Trends and spatial variation of oceanic dimethyl sulfide under a warming climate revealed by an artificial neural network model

Lyu Yan[1] and Wei-Lei Wang[1*]

[1]State Key Laboratory of Marine Environmental Science and College of Ocean and Earth Sciences, Xiamen University, Xiamen, China

Correspondence to: Wei-Lei Wang (weilei.wang@xmu.edu.cn)

**Abstract.** Marine dimethyl sulfide (DMS), a climatically active gas generated through microbial degradation of dimethyl sulfoniopropionate (DMSP), plays a key role in the Earth's climate system by modifying its radiation budget. However, the sea-to-air flux and future variations under climate change are still uncertain. Simulations from Earth System Models (ESMs) provide divergent trends. Here, we developed an artificial neural network (ANN) model trained using DMS observations and eight observational environmental parameters, along with model parameters extracted from the simulations of CESM2-WACCM to predict variations of DMS concentrations and sea-to-air flux for both historical (1850–2014) and SSP5-8.5 scenario (2015–2100). Our simulation indicates that DMS concentrations will generally decline by the end of this century. Specifically, from 2015 to 2050, the DMS concentrations are projected to decrease at a rate of 0.40±0.13% per decade. From 2050 to 2100, the rate of decrease is expected to accelerate to 0.89±0.08% per decade. The sea-to-air flux of DMS exhibits a non-monotonic trend. It is projected to increase at a rate of 0.51±0.16% per decade from 2015 to 2050. However, from 2050 to 2100, the flux is expected to decrease at a rate of 0.37±0.11% per decade. We further explore the attribution of DMS changes by running a series of sensitivity tests. We find that elevated sea surface temperature (SST) and photosynthetically active radiation (PAR), along with nutrient depletion, are projected to lead to the decline in DMS concentrations by the end of this century. Furthermore, our geospatial analysis indicates that mixed layer depth (MLD) emerges as the predominant driver in the Southern Ocean, and nutrient-dependent effects strongly correlate with DMS in the open seas (trades and westerlies). Our findings suggest that site-specific modeling schemes are needed to accurately model DMS dynamics.

## 1 Introduction

Oceanic dimethyl sulfide (DMS) is primarily synthesized in seawater through the enzymatic cleavage of the biogenic compound dimethyl sulfoniopropionate (DMSP) and released through microalgae exudation and mortality (Galí et al., 2015; Simó & Dachs, 2002; Stefels, 2000). DMS in the surface ocean is supersaturated compared to its atmospheric counterpart, and the sea-to-air flux is responsible for more than half of the total flux of gaseous sulfur to the atmosphere (Lana et al., 2011; Quinn & Bates, 2011). Once in the atmosphere, DMS is oxidized to sulfuric and methane-sulfonic acids, which contribute to the formation of cloud condensation nuclei (CCN) and facilitate cloud formation, and thereby has the ability to reduces solar radiation and affects the Earth's energy budget (Charlson et al., 1987). The CLAW hypothesis postulates a climate-negative feedback loop among phytoplankton, DMS emissions, CCN, and Earth's energy budget. In the proposed feedback loop, CCN and cloud albedo are regulated up or down by oceanic phytoplankton through the medium of DMS emissions (Charlson et al., 1987).

However, the conventional view of DMS induced negative climate feedback is increasingly challenged by emerging evidence. Both model simulations and mesocosm studies suggest that DMS may instead exert a positive climate feedback



effect under the global warming scenario (Six et al., 2013; Wang et al., 2018; Webb et
al., 2016; Zhao et al., 2024). $CO_2$ forcing simulations have been conducted to predict
DMS distribution and its global-scale emissions, with varying forcing conditions
revealing substantial spatial heterogeneity (Bopp et al., 2003; Gabric et al., 2013). As
a result of the combined effects of ocean acidification and climate change, the
projection of global DMS emissions decreases by about 18(±3)% in 2100 compared to
the pre-industrial time in an Earth system model (ESM) climate simulations (Six et
al., 2013). Similarly, global mean DMS concentrations are predicted to decrease by
15.1% by the end of this century compared to the historical results (1960-2014),
which is primarily driven by rising $CO_2$ levels (Zhao et al., 2024). In addition, a
declining trend in DMS concentrations and sea-to-air flux under the RCP8.5 scenario
are also detected using a fully coupled Earth system model (CESM) (Wang et al.,
62    2018).

An ensemble of four ESMs (CNRM-ESM2-1, MIROC-ES2L, NorESM2-LM, and
UKESM1-0-LL) from the CMIP6 historical and SSP5-8.5 experiments provided
divergent trends in DMS concentrations and sea-to-air flux starting from the year of
2015, which means that there is significant uncertainty regarding the changes of
projected DMS trends based on Earth System Models in the future (Bock et al., 2021).
A recent study by Joge et al. (2025) found that the global mean surface DMS
concentration exhibits a decreasing trend with warming. However, in contrast to the
decreasing trend of DMS concentration, the combined effects of increasing wind
speed, increasing sea surface temperature, and decreasing ice-coverage lead to an
increasing trend of sea-to-air DMS flux. Consensus has not been reached regarding
future trends in DMS concentrations and sea-to-air flux. Key factors controlling DMS
variations have yet to be determined. Consequently, the response of DMS
concentrations/flux under a warming climate remains uncertain.
Here, we train an artificial neural network (ANN) model using DMS measurements
(Fig. S1) and eight observational environmental variables, including Chl *a*, MLD,
dissolved inorganic nitrate (DIN), PAR, dissolved inorganic phosphorus (DIP),
silicate ($SiO_4$), sea surface salinity (SSS), and sea surface temperature (SST).
Subsequently, we feed the model with the monthly outputs from CESM2-WACCM
historical and SSP5-8.5 experiments to predict global DMS concentrations and
temporal variations (Fig. S2). Furthermore, we investigate the mechanisms driving the
variations of DMS concentrations under global warming scenarios. We also conduct
a series of sensitivity tests to explore the attribution of DMS changes in a warming
climate state, and identify the key factors (SST, PAR and nutrients) in different
regions. In the end, we make prospects and suggestions for future study.
**2 Data and Methods**
**2.1 Observations**
Observational DMS concentration data were obtained from two primary sources: 1)
the Global Surface Seawater DMS Database (Pacific Marine Environmental
Laboratory, PMEL; last access: 1 May 2020) and 2) the North Atlantic Aerosols and
Marine Ecosystems Study (NAAMES; Behrenfeld et al., 2019) (Table S1). After
quality control, which excluded measurements with DMS concentrations below 0.1
nM or exceeding 100 nM following Galí et al. (2015), a total of 93,571 valid data
points were retained (PMEL: 86,785; NAAMES: 6,786). The Global Surface
Seawater DMS Database also provides additional in situ measurements, including Chl
*a* (PMEL: 11,491; NAAMES: 6,750), SST (PMEL: 81,069; NAAMES: 6,786), and
SSS (PMEL: 77,209; NAAMES: 6,786). We used in situ measurements when they are
available. Otherwise, we supplemented the missing values with monthly climatology
data from auxiliary datasets (Table S1). For example, SeaWiFS monthly averaged
Level 3-binned Chl *a* data (9.2 km resolution; last access: 1 May 2020) from
December 1997 to March 2010 were spatially and temporally matched to DMS
measurements. Simillarly, SeaWiFS monthly averaged Level 3-binned PAR data (9.2
km resolution; last access: 1 May 2020) from September 1997 to August 2010 were



also matched with DMS observations. Climatological MLD data were downloaded
from the Monthly Isopycnal/Mixed-Layer Ocean Climatology (MIMOC; Schmidtko
et al., 2013). Nutrients (nitrate, phosphate, silicate) data were obtained from the World
Ocean Atlas 2013 (WOA2013). All ancillary data were aligned with DMS
measurements based on sampling location and time of year. Rigorous quality control
was applied following Galí et al. (2015). For instance, coastal data (salinity<30), as
well as measurements with anomalously low nutrient concentrations (phosphate<0.01
μM; nitrate<0.01 μM; silicate<0.1 μM) and low Chl $a$ (Chl $a$<0.01 mg m$^{-3}$) were
excluded to focus on open-ocean conditions.
**2.2 Earth System Models**
For the input data to our ANN model, we first used the surface and monthly
environmental outputs from CESM2-WACCM. This model ensemble was selected
because it demonstrates the best overall results among the CMIP6 ensembles when
compared to observational data (see Fig S6). We then compare our predicted DMS
concentrations and fluxes with the outputs from four ESMs in CMIP6 (CNRM-
ESM2-1, MIROC-ES2L, NorESM2-LM, UKESM1-0-LL). More detailed descriptions
of these four ESMs are provided below.
The oceanic components and their respective resolutions for the four ESMs are
detailed in Table S2, which includes ensemble numbers for both the historical (1850-
2014) and SSP5-8.5 (2015-2100) experiments. All datasets were downloaded from the
CMIP6 Earth System Grid Federation (ESGF) nodes. These ESMs simulate the main
large-scale features of the ocean circulation. Recent studies have also shown that these
models have improved simulations of MLD, a key driver for marine biogeochemistry
and marine DMS emissions (Seferian et al., 2020).
For the CNRM-ESM2-1 model, DMS concentrations are computed using the
biogeochemical model PISCES, coupled with the global ocean general circulation
model (OGCM) NEMO. The version of PISCES used, PISCESv2-gas, includes a
module for simulating the cycle of gases relevant to climate. DMS flux is calculated
using the parameterization of gas exchange coefficients of Wanninkhof (2014)
(Michou et al., 2020).
In the MIROC-ES2L model, DMS concentrations are calculated based on the Aranami
and Tsunogai (2004) parameterization, which links sea surface DMS concentrations to
MLD and Chl-$a$ concentration as follows:
$$\text{DMS} = \begin{cases} \frac{60.0}{MLD} & if\ \frac{Chl}{MLD} < 0.02 \\ 55.8 \cdot \left(\frac{Chl}{MLD}\right) + 0.6 & if\ \frac{Chl}{MLD} > 0.02 \end{cases},$$
in which MLD and Chl $a$ are simulated by OECO-v2, coupled in MIROC-ES2L
(Hajima et al., 2020). DMS flux is calculated using Aranami and Tsunogai (2004)
parameterization.
For the NorESM2-LM model, the biogeochemical model iHAMOCC is coupled in the
global OGCM BLOM to compute DMS concentrations, which is a function of
temperature and export production (Tjiputra et al., 2020).
For the UKESM1-0-LL model, DMS concentrations are computed within the ocean
biogeochemistry model MEDUSA (Yool et al., 2013) and interactively coupled with
the global OGCM NEMO. DMS concentrations are linearly correlates with a
composite variable that includes the logarithm of Chl $a$, light, and nutrients. DMS flux
is calculated according to the air-to-sea gas transfer scheme of Liss and Slater (1974).
DMS concentrations in the atmosphere are subsequently modified through a number
of gas-phase aerosol precursor reactions within the stratospheric and tropospheric
chemistry schemes of the UKESM1-0-LL model (Mulcahy et al., 2020).



### 2.3 Artificial neural network model

The ANN model is a branch of artificial intelligence (AI), which builts with a fully connected network of nodes and neurons. Each neuron has an activation function and is connected to other neurons by iteratively determined weights (Gardner & Dorling, 1998). This algorithm has a great advantage because they make no prior assumptions on the data distribution and can fit data in gap area using non-linear equation (Breiman, 2001; Gardner & Dorling, 1998).

The ANN model is trained using the *Keras* deep-learning toolbox in Python 3.8, with eight environmental variables (Chl *a*, MLD, DIN, DIP, PAR, SiO$_4$, SST, SSS) as predictants and DMS as predictor. All data are log transformed and normalized to the range of [−1,1].

The dataset is then divided into three sections: training, internal testing, and external validating datasets. Specifically, data falling into the following 14 latitude bands (64–65°N, 54–55°N, 44–45°N, 34–35°N, 24–25°N, 14–15°N, 4–5°N, 4–5°S, 14–15°S, 24–25°S, 34–35°S, 44–45°S, 54–55°S, 64–65°S) are left out for internal testing (9084 points). Similarly, data falling to the fifteen latitude bands (69–70°N, 59–60°N, 49–50°N, 39–40°N, 29–30°N, 19–20°N, 9–10°N, 1–0°S, 9–10°S, 19–20°S, 29–30°S, 39–40°S, 49–50°S, 59–60°S, 69–70°S) are left out for external validation (10870 points). The remaining data are used as training dataset (63042 data points). Separating the data by latitude bands rather than using random separation helps prevent information leakage, as in situ measurments are internally correlated. The traditional random separation methods tend to overfitting (Wang et al., 2020).

In the training process, we adjust the hyper-parameters, such as dropout ratio, number of hidden layers, and number of nodes on each layer to prevent overfitting while achieving the best goodness of fit to observations. Eventurally, the finial ANN model adopted consists of one input layer, two dense hidden layers, and one output layer. The input layer comprises nodes corresponding to the predictors. Each hidden layer contains 128 nodes, and the output layer has a single node for DMS concentration simulations. To mitigate overfitting, two dropout layers with a dropout ratio of 0.25 are incorporated into each hidden layer. Additionally, an L2 kernel regularizer with a value of 0.001 is applied to each hidden layer. During network training, the mean squared error of the internal validation data is monitored. After obtaining a satisfactory combination of those hyper-parameters, we fix them and finetune the network using all available data.

### 2.4 Sea-to-air flux of DMS

DMS flux is calculated using an empirical formula, which takes into account sea surface wind (SSW), sea ice coverage, and the viscosity coefficient related to gas transfer velocities in atmosphere and surface ocean.

Air–sea gas transfer is estimated using the following bulk formula:

$$F = K_w(C_w - C_a/H) \ ,$$

where $F$ is sea-to-air gas exchange flux, $C_w$ and $C_a$ are bulk water and gas concentrations, and $K_w$ $(cm \ h^{-1})$ is the overall gas transfer velocity, expressed in waterside units (Liss & Merlivat, 1986). $K_w$ reflects the combined resistance to gas transfer on both sides of the interface, as follows:

$$\frac{1}{K_w} = \frac{1}{k_w} + 1/Hk_a \ ,$$

where the dimensionless $H$ is the Henry law constant (gas or liquid), and $k_a$ and $k_w$ are gas transfer velocities in air and seawater, respectively. DMS in the surface ocean is strongly supersaturated with respect to that in the overlying atmosphere (Cw ≫ Ca), so the DMS flux bulk formula is simplified as:

$$F = K_w C_w$$





Our study uses the parameterization for $K_w$ that refer to Goddijn-Murphy et al.
(2012) (hereafter GM12), which is based on regressions between satellite-based wind
speed observations and shipboard in situ measurements of DMS gas transfer velocities
using eddy covariance method. SSW and sea ice area coverage data are from the
CESM-WACCM monthly simulation datasets.
**3. Results and discussion**
**3.1 Long-term trends of DMS under global warming scenario**
The ANN model captures the major variance in the observed data with the goodness-
of-fit $R^2$ value of 0.68 for the training datasets and 0.66 for the testing datasets (Fig.
S3). We then conduct temporal simulations by feeding the ANN model with
parameters extracted from CESM2-WACCM, which best reproduces the
corresponding observational parameters among CMIP6 ensembles (Fig. S4).
Compared to the historical pattern, the model reveals distinct trends in DMS
concentration across the global ocean, with notable patterns emerging in several key
areas (Fig.1a). For instance, DMS concentrations exhibit an increasing trend in the
Southern Ocean, the eastern equatorial Pacific, the subpolar North Atlantic, and the
Arctic Ocean, with the highest concentration increase occurring in the Southern
Ocean between 40°S and 60°S. This is particularly important because the Southern
Ocean is far from anthropogenic aerosol sources, and the sea-to-air flux of DMS is the
major source of atmospheric sulfur. Therefore, it strongly influences the radiative
budget in the Southern Hemisphere (Hamilton et al., 2014).
Nevertheless, a decreasing trend in DMS concentrations is evident in the low-to-
middle latitude regions of the Pacific, Indian, and Atlantic Oceans. Using a similar
network model, Joge et al. (2025) found that DMS concentrations increase in the
subtropical gyres, whereas we observe a decreasing trend in the same regions. The
discrepancy is primarily attributed to two factors: 1) Data sources: We trained our
model using predominantly observational parameters, while Joge et al. (2025) used
model outputs. Given the evident biases between model simulations and observational
data, we believe that the observational parameters are more effective in capturing the
true relationship with DMS concentrations than model-devrived data. 2) Model
Ensembles: We employed output solely from CESM2-WACCM as input to our
network model, whereas Joge et al. (2025) used an ensemble of eight models. As
illustrated in Fig. S4, CESM2-WACCM demonstrated the best reproduction of
observational data, while other models exhibit significant biases. These differences in
data sources and model ensembles likely account for the divergent results observed in
DMS concentration trends. The sea-to-air flux of DMS generally follows a similar
trend to DMS concentrations. The correlation sign is consistent in most of the open
oceans, except for regional discrepancies in coastal biomes. These discrapencies are
probably caused by the inverse change of wind speed with DMS concentrations.
To elaborate more on the trend of DMS with global warming, we calculate the global
area-weighted annual mean DMS concentrations and DMS flux. The temporal trend
from our ANN model is plotted alongside another neural network model and four
ESM ensembles, all of which explicitly model DMS under historical and SSP5-8.5
scenarios, spanning from 1850 to 2100 (Fig.2 and Table 1, 2). DMS concentrations
and fluxes in all models show a similar flat trend with different magnitudes over the
historical period. For future projections, our global mean surface DMS concentration
shows a decreasing trend, consistent with Joge et al. (2025). However, the global
mean sea-to-air flux of DMS exhibits a non-monotonic trend. From 2015 to 2050, the
sea-to-air flux of DMS shows a similar increasing trend to that reported by Joge et al.
(2025), but with a higher increasing rate (Table 2). This increasing trend is likely due
to the combined effects of decreasing ice coverage, increasing wind speed, and
increasing sea surface temperature, which compensate for the decreasing DMS
concentration (Fig. 1). However, from 2050 to 2100, the increasing trend reverses to a
decreasing trend, with a rate of 0.37±0.11% per decade. Among the CMIP6 models,
two EMSs (NorESM2-LM, and UKESM1-0-LL) predict decreasing trends in the





future, while the other two models (CNRM-ESM2-1 and MIROC-ES2L) predict the
opposite. This divergence has previously been suggested to be explained by the bias in
modelled SST (Bock et al., 2021).
**3.2 Attribution of DMS changes under global warming scenario**
To identify the parameter(s) driving the temporal variations of DMS, we conduct
eight sensitivity experiments. In each, we hold seven of the eight environmental
paramters needed by the ANN at their initial historial values, and allow the remaining
one to vary according to historical and SSP5-8.5 simulation. These eight experiments
are denoted as VChl, VMLD, VDIN, VPAR, VDIP, VSiO$_4$, VSSS, and VSST,
representing the varying parameter of Chl a, MLD, DIN, PAR, DIP ,SiO$_4$, SSS, and
SST, respectively. For the eight parameters, PAR and SST display an increase trend,
and the other six variables show a decrease trend under the SSP5-8.5.
DMS concentration shows a significant increase in the VSST test and a modest
increase in the VPAR test compared to the control run, which is consistent with the
trends of SST and PAR. The elevated SST, especially in high latitude oceans,
promotes phytoplankton production, which is the primary producer of DMS (del Valle
et al., 2007; Derevianko et al., 2009; Galí et al., 2013; Watanabe et al., 2007). VPAR
test reveals a modest positive correlation between the changing trends of DMS
concentrations and PAR (Fig. 3a, b).The higher irradiance inhibits bacterial
consumption of DMS, influencing the proportion of high DMSP producers within
assemblages (Galí et al., 2011; McNabb & Tortell, 2022; Vance et al., 2013).
Conversely, the distribution of PAR shows an overall negative spatial correlation with
DMS (Fig. 1a and Fig. S5), which may indicate a role for photolytic degradation in
DMS loss (del Valle et al., 2007). These findings suggest that light-induced oxidative
stress and inhibited microbial DMS consumption may influence regional DMS
distributions. This particularly true in areas where photolysis significantly drives
DMS oxidation. The relative contributions of biotic and abiotic processes require
further in situ validation.
DMS exhibits a decrasing trend in the VMLD test, likely because the fact that the
shoaling of MLD due to global warming inhibits the upwelling of bottom nutrients
(Fig.3a), hinders the vertical mixing of higher nutrients from deeper layers and
oxygen-rich waters in the upper ocean, suppressing phytoplankton primary production
in low-to-middle latitude oceans and ultimately resulting in a decline of DMS
concentrations in the surface ocean. In the VChl test, DMS concentration remains
nearly unchanged, which differs from the decreasing trend of global mean Chl-*a*
concentration. This is in contrast to previous studies that extensively link DMS to Chl
*a*, and indicates that the biogeochemical cycle of DMS is far more complex than Chl *a*
can represent (Galí & Simó, 2015; Nemcek et al., 2008; Simó & Dachs, 2002).
Indeed, applying an algorithm based on Chl *a* yielded little insight into DMS
dynamics (Hirata et al., 2011). This is probably bacause the taxonomic composition of
phytoplankton assemblages that differ in their ability of DMS production likely
influence the variability of DMS cycling. As such, the bulk Chl *a*, representing a
composite signal from all phytoplankton taxa, may have limited utility in predicting
the spatial patterns of DMS on a global scale, but may be useful regionally.
For the nutrient tests, DMS concentration show an increasing trend in both VDIN and
VDIP, while both DIN and DIP decrease under SSP5-8.5. Overall, DMS–nutrients
relationship may be partially attributed to the sulfur overflow hypothesis (Stefels,
2000), which suggests that nutrient-limited phytoplankton increase DMSP production,
and its subsequent cleavage to DMS as a mechanism to regulate intracellular sulfur
quotas when protein synthesis is limited (Hatton & Wilson, 2007; Kinsey et al., 2016;
Simó & Vila-Costa, 2006; Spiese & Tatarkov, 2014; Stefels, 2000). This hypothesis
also explains elevated DMS concentrations in the Southern Ocean, subpolar North
Atlantic and the Bering Sea, where nutrients concentrations are showing decreasing
trends (Fig. S5). Moreover, nutrient-dependent effects significantly explain seasonal



variability, particularly as phytoplankton growth becomes nutrient-limited during
summer time.
The ocean represents an intricate system where environmental changes directly
modulate DMS concentrations. Under the warming scenario (SSP5-8.5), elevated SST
and PAR will strengthen ocean stratification, shoaling the MLD and reducing nutrient
supplies from the deep ocean. Our findings suggest that these effects jointly
determines the temperol variation of DMS concentrations.

**3.3 Key factors regulating DMS variation in typical regions**

To further investigate these regional variabilities, we divide the ocean into six regions
according to Longhurst (1998): polar North, polar South, westerlies North, westerlies
South, trades, and coastal (see Fig. 4a). We then examine the key factors influencing
DMS concentrations across these regions (Fig. 4b). No key driving factors of DMS
variation are identified in the polar North and coastal regions, likely because these
areas encompass diverse biomes with site-specific drivers. When analyzed as a whole,
no single dominant factor emerges. In contrast, DMS concentrations in the westerlies
North region show strong negative correlations with DIN ($r = –0.46$), DIP ($r = –0.58$),
and $SiO_4$ ($r = –0.31$), and strong positive correlations with PAR ($r = 0.60$), SSS ($r =
0.52$), and SST ($r = 0.53$). These results suggest that low nutrient levels, strong light,
and warm surface waters may favor small phytoplankton or *Phaeocystis*, which are
more prolific DMS producers compared to diatoms.
Conversely, in the westerlies South region, Chl *a* and MLD emerge as the dominant
factors ($r = 0.44$ and $–0.48$, respectively) influencing DMS variation. This indicates
that Chl *a* is a strong predictor of DMS concentrations in this region, likely because
prolific DMS-producing phytoplankton contribute significantly to Chl-*a* levels. A
similarly strong negative correlation is observed between MLD and predicted DMS
concentrations in the polar South ($r = –0.48$). Both polar South and westerlies South
regions are characterized by high background nutrient concentrations and deep mixed
layers. The deepening of the MLD may dilute phytoplankton biomass and DMS,
leading to the observed relationships. In the trades region (open ocean), where
nutrient levels are generally low and small phytoplankton dominate, DMS
concentrations are positively correlated with DIN, DIP, PAR, and Chl *a*. This suggests
that in the trades, nutrient-driven higher primary production leads to higher DMS
production.
Observational and modeling studies have extensively documented the distribution of
DMS concentrations and sea-to-air fluxes across the global ocean (Galí et al., 2015;
Joge et al., 2025; Lana et al., 2011; Seferian et al., 2020; Simó et al., 2002). These
studies indicate that DMS emissions are not solely governed by global
biogeochemical cycles but also arise from complex ecological interactions, planktonic
food-web dynamics, cellular physiological processes, and marine chemical
transformations (Simó et al., 2002). Notably, elevated sea-to-air DMS fluxes are
predominantly observed in upwelling zones, particularly in the tropical and equatorial
Pacific Ocean. Furthermore, recent research highlights the influence of SST, deep-
water formation, biological productivity, and thermohaline circulation on DMS flux
variability (Seferian et al., 2020). Our results demonstrate that DMS concentrations
exhibit regional-scale dependence on multiple environmental drivers (Fig. 5).

**Conclusions**

The comparison of DMS concentrations and flux variations over the simulation
periods from 1850 to 2100 (historical and SSP5-8.5 for CMIP6 ESMs) yields two key
insights. Firstly, all models exhibit relative stability during the historical period. In
contrast, in future simulations, two models (CNRM-ESM2-1 and MIROC-ES2L)
show an increase in surface ocean DMS concentrations and flux, while the other four
models (Joge25, NorESM2-LM, UKESM1-0-LL, and ANN) show a decreasing trend
in DMS concentration. Although ESMs or non-linear equations may not fully





elucidate the relationship between DMS and marine phytoplankton, clarifying its
response to climate change is crucial.
Secondly, our findings suggest that DMS concentrations exhibit regional-scale
dependence on multiple environmental drivers. In the trades region (open ocean),
higher DMS production is primarily driven by nutrient-mediated increases in primary
productivity. Conversely, in the westerlies North region, DMS concentrations display
strong negative correlations with DIN, DIP, and $SiO_4$, while showing strong positive
correlations with PAR, SSS, and SST. In the westerlies South region, Chl a emerges as
a key positive predictor of DMS, and MLD is negatively correlated with DMS in the
polar South.
Our results demonstrate that variations in DMS concentration are rarely unidirectional
in response to isolated changes in a single environmental parameter (Fig. 5). This
highlights the complex interactions among these environmental factors, which cannot
be adequately captured by a linear regression model. Future work should focus on the
combined effects, using observational data to constrain models, and integrating these
with ESMs to more accurately simulate DMS concentrations under different
scenarios. It is also crucial to consider the potential climatic implications of changes
in DMS production driven by biogeochemical factors when projecting future climate
change.
**Author Contributions**
W.-L. W. conceived the project. L.Y. and W.-L. W.carried out the formal analyses. Y.
L. and W.-L. W. wrote and reviewed the manuscript. Both authors have given
approval to the final version of the manuscript.
**Acknowledgements**
We thank the observational DMS community for making their measurements publicly
available. We also thank the authors and agencies for providing the ancillary data used
in this study. W.-L.W and Y.L. were supported by the National Natural Science
Foundation of China (42476031), and the Natural Science Foundation of Fujian
Province of China 2023J02001.
**Competing interests**
The authors declare that there are no competing interests.



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




**Table 1.** Summary of DMS trends for historical and future scenarios at different
durations from 1850 to 2100 under the high emission scenario (SSP5.85). The trends
were calculated according to the method described by Joge et al. (2025) using a
bootstrap approach. The unit is % per decade, which indicates the relative changes in
DMS concentration compared to the initial year of each period.

| | Trend $\pm$ SD % decade$^{-1}$ | | | | |
|---|---|---|---|---|---|
| **Model** | **1850-1900** | **1900-1950** | **1950-2014** | **2015-2050** | **2050-2100** |
| **CNRM-ESE2-1** | -0.06±0.01 | 0.02±0.01 | 0.06±0.01 | 0.41±0.03 | 1.19±0.04 |
| **MIROC-ES2L** | -0.01±0.02 | 0.15±0.02 | 0.43±0.02 | 0.82±0.04 | 0.69±0.02 |
| **NorESM2-MM** | 0.02±0.04 | -0.09±0.03 | 0.06±0.03 | -0.69±0.13 | -1.01±0.09 |
| **UKESM1-0-LL** | 0.04±0.03 | -0.09±0.03 | -0.18±0.03 | -1.51±0.08 | -1.26±0.06 |
| **Joge25** | 0.07±0.02 | 0.01±0.02 | 0.06±0.02 | -0.46±0.03 | -0.58±0.02 |
| **this study** | 0.14±0.08 | -0.12±0.09 | -0.45±0.06 | -0.40±0.13 | -0.89±0.08 |







**Table 2.** Summary of DMS flux trends for historical and future scenarios at different durations from 1850 to 2100 under the high emission scenario (SSP5.85). The trends are calculated according to the method described by Joge et al. (2025) using a bootstrap method. The unit is % per decade, which indicates the relative changes in DMS flux compared to the initial year of each period.

| | Trend $\pm$ SD % decade$^{-1}$ | | | | |
|---|---|---|---|---|---|
| **Model** | 1850-1900 | 1900-1950 | 1950-2014 | 2015-2050 | 2050-2100 |
| **CNRM-ESE2-1** | -0.04$\pm$0.05 | 0.03$\pm$0.04 | 0.37$\pm$0.03 | 0.68$\pm$0.13 | 1.32$\pm$0.07 |
| **MIROC-ES2L** | -0.005$\pm$0.05 | 0.18$\pm$0.03 | 0.33$\pm$0.04 | 1.16$\pm$0.06 | 1.45$\pm$0.03 |
| **NorESM2-MM** | -0.009$\pm$0.06 | -0.13$\pm$0.04 | 0.06$\pm$0.04 | -0.41$\pm$0.13 | -0.51$\pm$0.08 |
| **UKESM1-0-LL** | 0.03$\pm$0.04 | -0.06$\pm$0.03 | 0.17$\pm$0.02 | -0.71$\pm$0.09 | 0.03$\pm$0.06 |
| **Joge25** | 0.04$\pm$0.03 | 0.08$\pm$0.02 | 0.26$\pm$0.02 | 0.16$\pm$0.03 | 0.37$\pm$0.03 |
| **this study** | 0.01$\pm$0.09 | -0.04$\pm$0.07 | -0.07$\pm$0.06 | 0.51$\pm$0.16 | -0.37$\pm$0.11 |








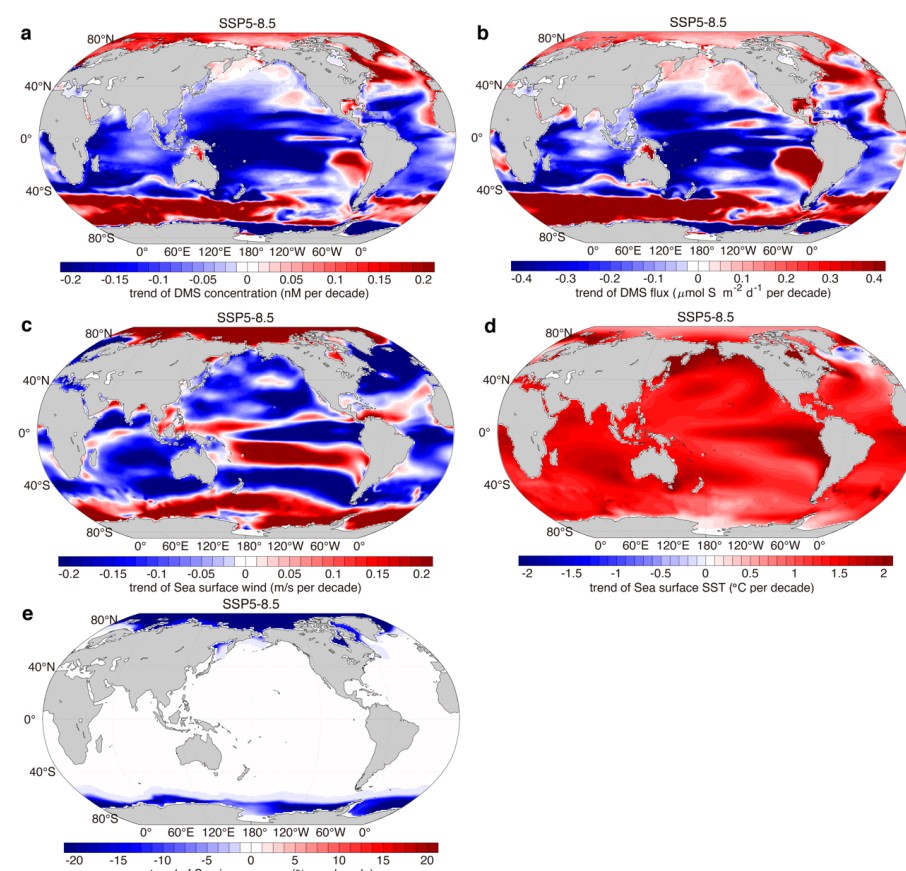

**Figure 1.** Changing trends of concentration and flux DMS with assimilatory
parameters from 2015 to 2100 under SSP5–8.5 simulation. **a,** trend of DMS
concentration. **b,** trends of DMS flux. **c,** trend of sea surface wind speed. **d,** trend of
sea surface temperature. **e,** trend of sea ice coverage.







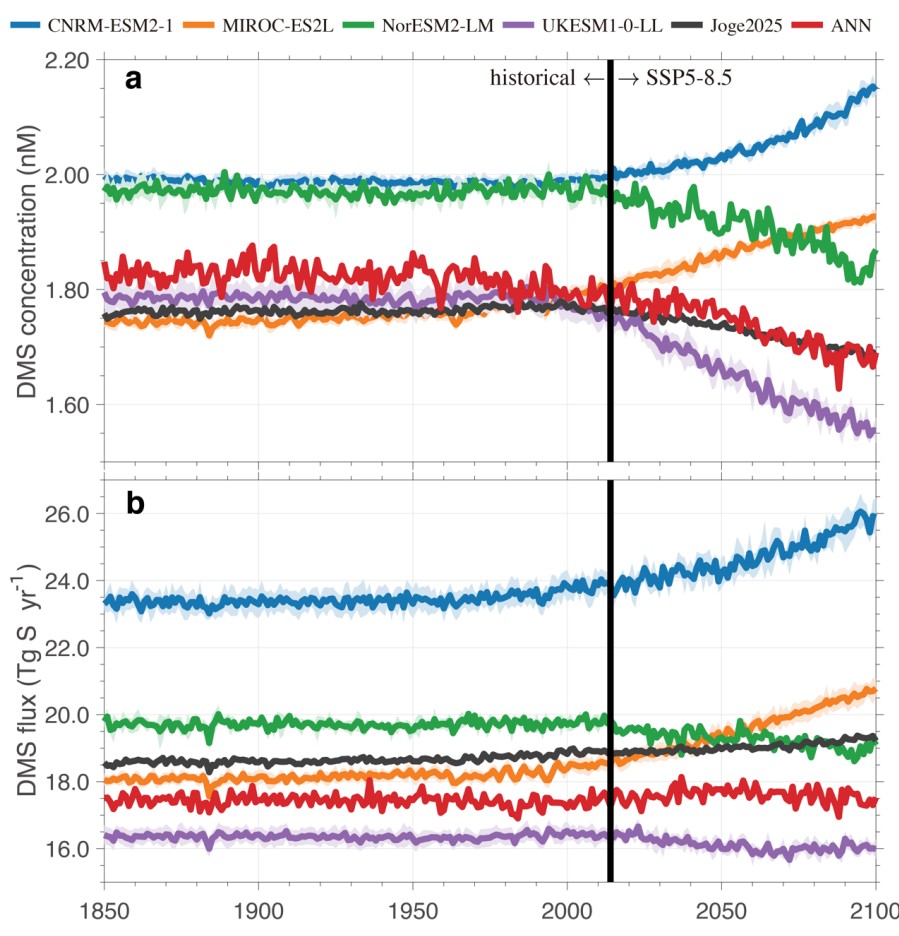


**Figure 2.** Time series of mean annual global area-weighted surface ocean DMS concentration and DMS flux over 1850-2100 (CMIP6 historical and SSP5-8.5 simulations). **a,** DMS concentration (nM). **b,** DMS flux (Tg S yr$^{-1}$)







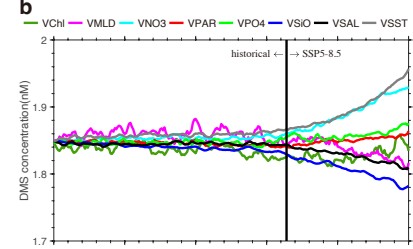


**Figure 3.** Time series of input variables and DMS concentrations of sensitivity tests
over 1850–2100. **a**, Time series of eight input environmental variables normalized to
(-1,1). **b**, Time series of mean annual global area-weighted DMS concentrations of
eight sensitivity tests.

605



606

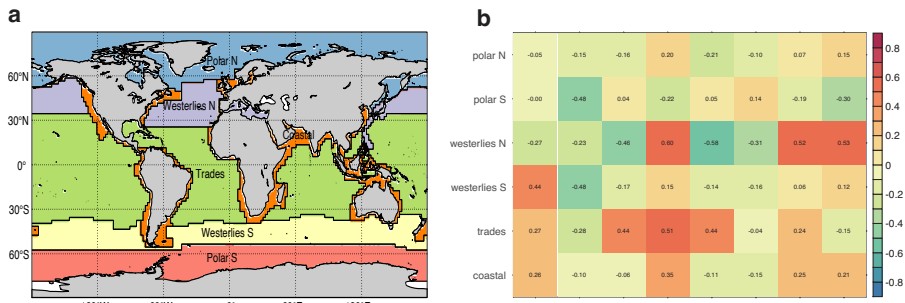

607

**Figure 4.** Correlation of DMS concentrations with environmental variables in six main regions. **a**, Six oceanic regions that were separated based on Longhurst's biomes (Longhurst, 1998). **b**, Correlation of DMS concentrations with eight input environmental variables in six oceanic regions.






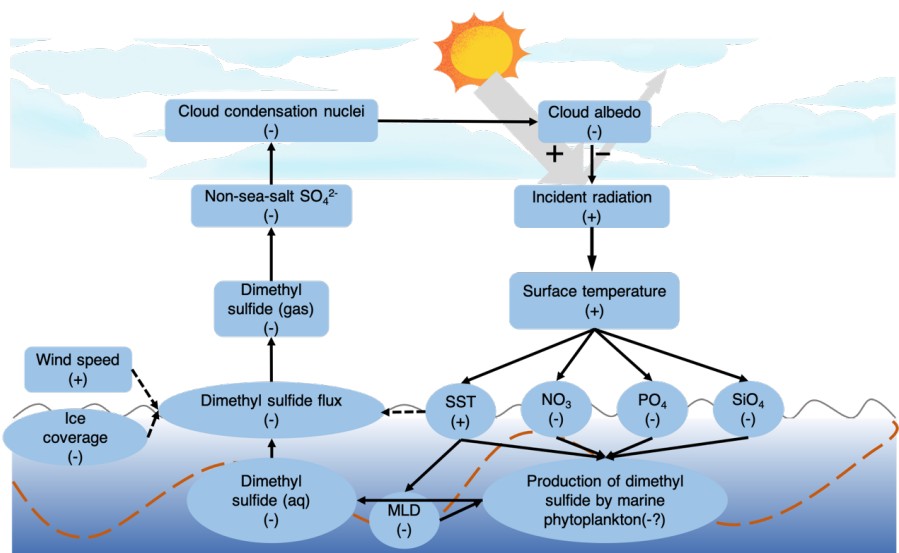


**Figure 5.** Modified diagram of the climate feedback loop of DMS.