# Peer review of "Trends and spatial variation of oceanic dimethyl sulfide under a warming climate revealed by an artificial neural network model"

_EGUsphere, 2025_

## Referee Comment (RC1)

Air-sea gas exchange is a fundamental process in the Earth system. Dimethyl sulfide (DMS) acts as a potential climate-active gas, and its air-sea flux plays an important role in climate regulation. Understanding how surface ocean DMS concentrations and fluxes respond to future climate change is therefore essential. This study applies a novel neural network approach to investigate historical and future DMS distributions under a specific climate change scenario. Although similar studies exist, this work improves data selection for both training and projection phases and produces some distinct results. The drivers of DMS variability are further examined through sensitivity tests. The paper is well written, the topic is timely, and the method is appropriate. It could be a valuable contribution to the community after addressing the following comments.

Line 16 – I am not a climate modeling expert, but why was SSP5-8.5 specifically chosen? Please clarify the rationale.

Line 116 – Consider explaining the full name of CESM2-WACCM and adding a reference.

Line 117 – Possibly a naive suggestion, but did you consider using the best-performing variables from different models? For example, IPSL-CM6A-LR performs well for $SiO_4$—can it be used selectively?

Line 118 – There is no Fig. S6; this likely refers to Fig. S4.

Table S2 – Do the last two columns represent the number of components? Please clarify.

Line 132 – A reference is needed to support this sentence.

Line 141 – If interpreted correctly, Aranami and Tsunogai (2004) did not introduce a new parameterization but used the Nightingale et al. (2000) formulation.

Line 142 – Please specify which gas exchange parameterization is used in this model.

Line 149 – The Liss parameterization differs substantially from Wanninkhof (2014) and Nightingale et al. (2000). This likely causes significant differences in DMS flux estimation.

Line 154 – A link to Fig. S2 would be appropriate here.

Line 174 – Wang et al. (2020) is not listed in the bibliography.

Line 211 – The value 0.6673 should be approximated as 0.67, not 0.66, for the testing/validation datasets.

Lines 224–241 – It would be informative to show the absolute concentrations and fluxes in both historical and projected periods. This would help assess whether high or low DMS regions are increasing or decreasing. Consider including these our subplots near Fig. 1 or in the Supplement.

Fig. 2 – It is notable that both this study and Joge et al. (2025) use neural networks, yet they predict systematically different historical DMS concentrations. Both presumably rely on the same observed DMS data for training. Even with different training variables, similar observed values should be reproduced, right? Please comment on this discrepancy.

Table 2 and Fig. 2 – Although the focus is on concentration, flux is also presented and highlighted in the abstract. Therefore, some clarification is needed. What gas transfer velocity (K) formulation is used in Joge et al. (2025)? That study predicts lower DMS concentrations but higher fluxes. Is this due to a

different K? The same issue may apply to the green and blue models in Fig. 2. Flux comparisons should use a consistent K to be comparable. Since wind speed modulates flux trends, the choice of K is likely significant. A discussion of this would be valuable near the flux results.

Lines 286–291 – This section requires supporting references.

Section 3.3 – The first two paragraphs offer plausible explanations but would be more convincing if supported by literature. The last paragraph includes many citations, consider distributing some of these to earlier parts of the section.

Fig. 5 – This is a strong figure but does not reach its full potential, which is a shame. I think captions should emphasize the key message, and the main text should include dedicated discussion for this figure.

Data availability – The journal likely requires a data sharing statement. Please check if the data repository link is missing.

-Yuanxu Dong

---

## Referee Comment (RC2)

**1 General comments**

Yan and Wang (hereafter YW25) present a study of oceanic dimethyl sulfide (DMS) modeled by an artificial neural network (ANN) over historical (1850–2014) and SSP5-8.5 (2015–2100) CMIP6 scenario. DMS is the main natural source of sulfur in the atmosphere, which contributes to aerosol formation, and ultimately affects the Earth radiative forcing, and thus the climate. Previous modeling studies have shown large discrepancies in the projected evolution of DMS. Thus, developing robust methods to model DMS is of high importance to reduce the uncertainty in climate models.

Despite the importance of this subject, the study proposed by YW25 presents major issues. In this section, general comments are presented, and more details are provided in the Specific Comments section for some of them.

The primary concern with YW25 paper is about scientific integrity. The ANN used in this study was developed 5 years ago by Wang et al. (2020, hereafter referred to as W20), and one can thus expect the method presented here to be essentially identical to that of W20. However, one would also expect the entire process to be revisited and updated if needed, for instance with new input data that has been published since 2020. Instead, several parts of this paper have simply been duplicated from W20. The original study is not acknowledged, and the reference to W20 is not even present in the bibliography. More concerning is that a whole subsection of the paper, two figures, and a table strongly resemble published material from other authors. This will be presented with more details in the Specific comments section of this review.

Another major issue (which is partly linked to the previous remark) is that this work lacks novelty. It is similar to existing work, and does not provide significant new insights. The recent study by Joge et al. (2025, hereafter referred to as J25) uses a similar modeling technique (but is trained on more complete observational datasets), and provides a more thorough investigation of several climatic scenarios (only one of which is evaluated here). J25 uses forcing data from a multimodel ensemble (instead of a single model here). While an interesting comparison could be made to identify and address the differences between the two studies, the analysis of the respective results is weak and limited. For example, the differences in concentration and flux during the historical period are not mentioned in the paper, even though it is surprising that the ANN computes a larger concentration but a smaller flux than the J25 model. The diverging trend in DMS flux between the two studies for the 2050–2100 period (+0.37 and −0.37 % per decade in J25 and this study, respectively, see Table 2) is not clearly established in the paper despite being one of the most important expected result from such a study.

Several methodological aspects are vague, or even flawed. This is especially the case regarding the choice of a single model (CESM2-WACCM) to provide input data for the ANN. The paper states that CESM2-WACCM is the best CMIP6 model for eight variables, but some of the provided data suggest otherwise. The evaluation of uncertainty range is neither explained, nor discussed (despite being significantly larger than in other studies).

Lastly, the presentation of the paper is poor. There are missing references and inconsistencies throughout the text. Some statements attributed to other studies are wrong. Several figures are barely readable. A number of misspelled words reveal that the paper was not even spell-checked before submission.

In summary, this paper has major weaknesses and lacks scientific rigor. In my opinion, it does not meet the quality standards required for publication in Earth System Dynamics.

**2  Specific comments**

**2.1  Copies from other papers**

Section 2.1 (Observations) is largely copied and pasted from W20. The last access date (May 1, 2020) has not been updated. Although it is acceptable to reuse the same method as in the previous study by Wang et al. (2020), this should be clearly stated and referenced. It would have been expected to use an updated dataset of DMS measurements. Note that two references (Behrenfeld et al., 2019; Schmidtko et al., 2013) are also missing from the bibliography: again, no adjustments were made after copying.

Section 2.2 (Earth System Models): apart from the first two paragraphs, each CMIP6 model description is copied from Bock et al. (2021, hereafter B21), with less information (see Table R2 for a side-by-side comparison).

In Section 2.3 (Artificial neural network model), paragraphs 2 and 3 are mostly copied from W20. Using the same method/model is acceptable, but this should be explicitly acknowledged.

Section 2.4 (Sea-to-air flux of DMS; lines 191–206) is an exact (but partial) copy of W20. As previously mentioned, the text from W20 was not adapted: for example, the abbreviation "GM12" for the Goddijn-Murphy et al. (2012) paper has been copied from W20 but is never used throughout this paper.

Figure 2 strongly resembles Figure 9 from B21. Time series from J25 and the present study have been added to it. However, given the resemblance, one would expect to see a mention of "adapted from" and the source. In this figure, the solid lines are too thick, and the envelopes surrounding the solid lines are not explained in the caption. They also seem different from those displayed in B21 Fig. 9. Why is that? The data from J25 is incomplete since it does not include the uncertainty range provided in their study. This range is very important information for such a study.

Figure 5 strongly resembles Figure 1 from Quinn and Bates (2011, hereafter referred to as QB11), which discusses the CLAW hypothesis after the publication of Charlson et al. (1987). Both are shown in Appendix (Fig. R1). The caption of Fig. 5 is a copy of the caption from QB11 ("Modified diagram of the climate feedback loop of DMS"), but QB11 clearly states the reference from which it was "modified" (their caption continues by: "[. . .] feedback loop proposed by ref. 7"). Additionally, the caption of Fig. 5 is limited to a title. An explanation of the sketch, including the significance of the plus and minus signs, should have been provided. Only five out of the eight variables (SST, MLD, $NO_3$, $PO_4$, $SiO_4$) are displayed, while the other three (Chl a, SSS, PAR) are not displayed, without explanation or justification. The authors attribute a sign to the effect of these changing variables on DMS; however, this contradicts their own conclusions (line 375) "Our results demonstrate that variations in DMS concentration are rarely unidirectional in response to isolated changes in a single environmental parameter (Fig. 5)."

Fig S3b is an exact copy of Figure 1b from W20. Note the rounding error for $R^2$ (0.6673 in Fig S3b, rounded as 0.66 in the text, lin 211), as well as the inconsistency with N values (lines 170–171).

Table S1 is nearly identical to Table A1 in W20. Two out of four URLs are no longer valid. The "last access" dates have not been updated since W20 ("1 May 2020"). The Table caption has been modified compared to W20 but contains a mistake ("DMS and environmental paraments data sources"). All the references provided in this Table are missing from the bibliography, further evidencing that no adjustments were made after copying from the W20 paper.

Table S2 is a copy of two tables in B21 (Table 1 for the three columns on the left and Table 2 for the 2two columns on the right). The copy is obvious in this table since the number of CMIP6

model realizations displayed in the last two columns is no longer up to date (more realizations are now available for almost all models and scenarios compared to 2021).

**2.2 Justification of CESM2-WACCM choice, use of CMIP6 models**

In several places in their paper (lines 116–118, 213–214, and 235–236), YW25 claim that CEMS2-WACCM outperforms other CMIP6 models in reproducing the eight environmental variables used as input in the ANN. They refer to Figure S4 (and mistakenly to Figure S6, which does not exist; line 118) to support this statement. However, Figure S4 does not demonstrate that CESM2-WACCM is superior.

Figure S4 presents two metrics to evaluate model performance against measurements: $R^2$ values, and a percentile in color bar. The second metric is not explained in the "Methods" section of the paper and is nearly impossible to read in the figure, even on a large screen. Using only the reported $R^2$ values for the eight variables and computing the mean $R^2$ over eight variables for seven models, it appears that CESM2-WACCM is only the third-best model (Table R1).

| Model | Chl a | MLD | DIN | PAR | DIP | $SiO_4$ | SSS | SST | mean | rank |
|-------|-------|-----|-----|-----|-----|---------|-----|-----|------|------|
| CESM2 | 0.2163 | 0.2317 | 0.9097 | 0.5316 | 0.8797 | 0.7826 | 0.8448 | 0.9465 | 0.668 | 4 |
| CESM2-WACCM | 0.2519 | 0.2616 | 0.9109 | 0.5345 | 0.8836 | 0.7638 | 0.8402 | 0.9470 | 0.674 | 3 |
| NorESM2-LM | 0.2163 | 0.1022 | 0.7570 | 0.5319 | 0.7805 | 0.6606 | 0.8980 | 0.9370 | 0.610 | 6 |
| NorESM2-MM | 0.2682 | 0.1125 | 0.7074 | 0.5304 | 0.7692 | 0.6206 | 0.9054 | 0.9421 | 0.607 | 7 |
| EC-Earth3-CC | 0.3206 | 0.2621 | 0.9103 | 0.5177 | 0.7961 | 0.7982 | 0.8570 | 0.9345 | 0.675 | 2 |
| GFDL- ESM4 | 0.5607 | 0.1829 | 0.8525 | 0.5236 | 0.8372 | 0.8664 | 0.8147 | 0.9470 | 0.698 | 1 |
| IPSL-CM6A-LR | 0.2210 | 0.2131 | 0.9032 | 0.5066 | 0.7899 | 0.8867 | 0.8052 | 0.9429 | 0.659 | 5 |

Table R1: $R^2$ values provided in Figure S4. Here, the mean over eight variables is computed for each model, and the resulting rank is provided. CESM2-WACCM ranks third.

The claim that CESM2-WACCM is superior is not supported and appears flawed. The authors also state that "other models exhibit significant biases" (line 236), but this is clearly not true, considering the relatively narrow range of correlation coefficients for all models, including CESM2-WACCM (Table R1).

More generally, the authors' choices regarding the presentation and selection of the CMIP6 models are questionable. In Section 2.2, CESM2-WACCM is presented as "the best [. . . ] among the CMIP6 ensembles" (line 117) without being described. Then, the four CMIP6 models that compute DMS concentration and flux are presented in great detail from lines 129 to 152. However, none of this information is used later in the paper to explain or analyze the differences in the results. What is the purpose of this extended model description? The same question can be asked about Table S2, which also ignores CESM2-WACCM; the information in this table is never referenced in the paper.

Regarding Figure S4, which supposedly justifies why CESM2-WACCM is better, three out of the four CMIP6 models presented in detail in Section 2.2 (CNRM-ESM2-1, MIROC-ESM2-L, and UKESM1-0-LL) are missing. Why were these models excluded from the comparison? Conversely, what led to the selection of these seven models? The model data is not clearly stated either (which, and how many simulations are presented on this Figure?). What justifies the selection of two pairs of similar models (CESM2 and CESM2-WACCM, which differ in their atmospheric chemistry scheme; and NorESM2, which has two distinct resolutions, LM and MM)?

Last but not least, one strength of climate model simulations is the range of variability that

can be assessed from multiple realizations. However, it is unclear whether the authors used one or multiple realizations of CESM2-WACCM in their study. No information is provided regarding the number of available simulations (although this information is provided for other models in Table S2). Tables 1 and 2 provide uncertainty ranges without explanation of how they were evaluated. Conversely, Figure 2 shows only the mean global average, not the uncertainty. This clearly reduces the value of the study and limit its comparability with other studies.

**2.3   Other specific comments**

(23, 256) "decrease at a rate of 0.37 [...]". A decrease rate should be written with a minus sign.
(42) Presenting the CLAW hypothesis in 2025 is not appropriate, since this early study has been extensively revisited (see, for instance, Woodhouse et al., 2010, SB11, or Brévière et al., 2015). Stating that this hypothesis "is increasingly challenged by emerging evidence" (l. 47) is inadequate.
(173) Refer to the a specific paper demonstrating the overfitting of random separation method. This is mentioned, but not demonstrated by W20.
(188) "DMS flux is calculated using an empirical formula, which takes into account sea surface wind (SSW), sea ice coverage, and the viscosity coefficient related to gas transfer velocities in atmosphere and surface ocean." This description is not consistent with the formulas presented later (lines 192 and 202), which do not include sea surface wind (at least, not explicitly) or sea ice coverage. The flux calculation description provided in this section is not precise or complete enough to reproduce the results.
(222) YW25 comment about DMS concentration here, but refer to the study by Hamilton et al. (2014) which deals with CCN in pristine environments. The relative contribution of DMS to CCN is not established here and has been shown to be small by Woodhouse et al. (2010). Furthermore, Hamilton et al. (2014) points out that sea spray is the dominant source of aerosol, which contradicts your point: "A Southern Ocean summertime band of pristine CCN exists between 50°S and 65°S, with generally low monthly mean PD CCN concentrations of 20–153 $cm^{-3}$ (median 58 $cm^{-3}$), when natural emissions of sea spray are the dominant aerosol source (17)." (p. 18468)
(228–234) The analysis of discrepancies with J25 results is weak, and fails to address the main arguments: the model developed by J25 is specifically trained and applied over four biomes. Then, the results are combined to obtain global DMS concentrations (J25, Fig. 1). J25 also use the most recent database of DMS, which contains 853,343 filtered data points, while YW25 use a more limited dataset of 93,571 filtered data points. Consequently, in J25 study, DMS measurements are captured with significantly higher correlation coefficient (R=0.86; J25 p. 2) than with the ANN, which was randomly trained and tested over 14 and 15 (respectively) latitude bands encompassing all biomes.
(259) There is no such conclusion in B21, who wrote (only about the DMS flux): "A bias in modelled SST can thus contribute to the bias in flux calculation, but it is estimated to be smaller than the uncertainty of flux parameterisation." (p. 3836). This does not refer to the divergence of CMIP6 models. Additionally, the purpose of this study is not to evaluate differences in CMIP6 models, so this (wrong) citation is outside the scope of the paper.
(264) "at their initial histori[c]al values": an accurate description of the sensitivity experiment should be provided in the Methods section. The initial historical value is not explicit. Is it the first timestep, the first annual mean, or the mean over a longer period?
(311) "Moreover, nutrient-dependent effects significantly explain seasonal variability, particularly as phytoplankton growth becomes nutrient-limited during summer time." This is not supported by your work, add a citation.

(314–317) add a citation

(323) "No key driving factors of DMS variation are identified in the polar North and coastal regions, likely because these areas encompass diverse biomes with site-specific drivers." The precise meaning, and associated geographical extent, of the term "biome" should probably be clarified, as different authors use it in different ways. For example, J25 divided the global ocean into four "biomes": polar, trades, westerlies, and coastal. But whatever the word used, in this part of the study, the sensitivity of DMS is analyzed in six "regions". For two of these regions, which show weak correlation with all environmental variables used as input, the authors justify this by saying that these regions "encompass diverse biomes". Yet the other four regions surely also encompass diverse biomes and present stronger correlations. Therefore, this cannot be a valid argument explaining why no key driving factors stand out in the Polar North and coastal regions. This is also inconsistent with other statements in the paper. For instance, lines 329–331 state: "These results suggest that low nutrient levels, strong light, and warm surface waters may favor small phytoplankton or *Phaeocystis*, which are more prolific DMS producers compared to diatoms." These statements attempt to demonstrate that the ANN could capture the underlying biological processes responsible for DMS production. If the ANN can do so in some regions, why would it fail in others?

(326) "In contrast, DMS concentrations in the westerlies North region show strong negative correlations with DIN (r = −0.46), DIP (r = −0.58), and $SiO_4$ (r = −0.31), and strong positive correlations with PAR (r = 0.60), SSS (r = 0.52), and SST (r = 0.53)." It is unbelievable that an r value as low as 0.31 (and not higher than 0.60) would be considered a "strong correlation" in a scientific paper.

(326) These results should be compared with the available literature, McNabb and Tortell (2022) for instance.

(329) "[...] small phytoplankton or *Phaeocystis*, which are more prolific DMS producers compared to diatoms." This is not supported by your work, add a citation.

(345–355) This whole paragraph is not really a discussion about the ANN results, and would rather be in the Introduction.

(356, 367) "Our results demonstrate that DMS concentrations exhibit regional-scale dependence on multiple environmental drivers (Fig. 5)." This is not a new result: W20 already showed in their sensitivity tests that changes in DMS concentration exhibit strong regional variations (including opposite signs) when the drivers are perturbed individually (Fig. 9 in W20).

(571) In Table 1 and 2, uncertainty ranges (standard deviation) are provided. The first question, mentioned earlier, is: where does it come from? Is it based on the use of several model realizations for the input data? From both Tables, it appears that the ANN uncertainty is significantly higher than in all other studies, why is that?

**3 Technical corrections**

Here, I report only a short list of blatant mistakes, that would have been avoided if a spell-check had been performed before submitting.

- (103) "Simillarly"

- (173) "measurments"

- (185) "finetune"

- (232) "devrived"

- (240) "discrapencies"

- (260) "modelled"

- (264) "paramters"

- (264) "historial"

- (286) "decrasing"

- (292) "differes"

- (297) "bacause"

- (318) "temperol"

- (320) "variabilities"

- (Table S1) "paraments"

**4 Appendix**

Table R2: Side-by-side comparison of CMIP6 model description by B21 (left column) and YW25 (right column) showing strikingly strong resemblance.

| B21 (Sect. 2.1.1 pages 3825–3826) | YW25 (Sect. 2.2 lines 129–152) |
|---|---|
| In CNRM-ESM2-1, DMS concentration is computed by the biogeochemical model PISCES (Aumont and Bopp, 2006), | For the CNRM-ESM2-1 model, DMS concentrations are computed using the biogeochemical model PISCES, |
| embedded within the global general ocean circulation model NEMO. [. . . ] | coupled with the global ocean general circulation model (OGCM) NEMO. |
| The version of PISCES used in CNRM-ESM2-1 for CMIP6 is PISCESv2-gas, based on PISCES-v2 (Aumont et al., 2015) | The version of PISCES used, PISCESv2-gas, |
| with the addition of a specific module to compute | includes a module for simulating |
| the cycle of gases relevant to climate. [. . . ] | the cycle of gases relevant to climate. |
| The fluxes to the atmosphere are then | DMS flux is |
| computed using the parameterisation of gas exchange coefficients of Wanninkhof (2014). [. . . ] | calculated using the parameterization of gas exchange coefficients of Wanninkhof (2014) |
| (see Michou et al., 2020, for details). | (Michou et al., 2020). |
| NorESM2-LM includes a fully interactive description of the DMS cycle [. . . ] | For the NorESM2-LM model, [. . . ] |
| DMS is directly released in the water and is computed as | DMS concentrations, which is |
| a function of temperature and simulated detritus | a function of temperature and |

| | |
|---|---|
| export production (Tjiputra et al., 2020). | export production (Tjiputra et al., 2020). |
| In MIROC-ES2L, the seawater concentration of DMS is computed according to the parameterisation of Aranami and Tsunogai (2004), which relates the sea surface DMS concentration to the MLD and to surface water Chl concentration. [. . .] $$\mathrm{DMS} = \begin{cases} 60.0/\mathrm{MLD} & \text{if Chl/MLD} < 0.02\,\mathrm{mg\,m^{-4}} \\ 55.8 \cdot (\mathrm{Chl/MLD}) \\ \quad +0.6 & \text{if Chl/MLD} \geq 0.02\,\mathrm{mg\,m^{-4}} \end{cases}$$ Both MLD and Chl are simulated by the ocean biogeochemical model OECO-v2 embedded in MIROC-ES2L (Hajima et al., 2020). [. . .] The flux of DMS to the atmosphere is also computed according to Aranami and Tsunogai (2004). | In the MIROC-ES2L model, DMS concentrations are calculated based on the Aranami and Tsunogai (2004) parameterization, which links sea surface DMS concentrations to MLD and Chl-a concentration as follows: $$\mathrm{DMS} = \begin{cases} \frac{60.0}{MLD} & if\ \frac{Chl}{MLD} < 0.02 \\ 55.8 \cdot \left(\frac{Chl}{MLD}\right) + 0.6 & if\ \frac{Chl}{MLD} > 0.02 \end{cases}$$ in which MLD and Chl a are simulated by OECO-v2, coupled in MIROC-ES2L (Hajima et al., 2020). DMS flux is calculated using Aranami and Tsunogai (2004) parameterization. |
| In UKESM1-0-LL, the seawater concentration of DMS is computed within the ocean biogeochemistry model MEDUSA (Yool et al., 2013) and is interactively coupled with the atmosphere. The parameterisation of DMS concentration is based on the work by Anderson et al. (2001) and linearly relates the DMS concentration to a composite variable formed by the logarithm of the product of Chl concentration (C, mg m$^{-3}$ ), light (J , mean daily shortwave, W m$^{-2}$ ), and a nutrient term (Q, dimensionless) that depends on nitrate concentration. [. . .] Finally, the flux of DMS from the surface ocean to the atmosphere is parameterised according to the air–sea gas transfer scheme of Liss and Merlivat (1986)[1]. DMS concentration in the atmosphere is subsequently modified through a number of gas-phase aerosol precursor reactions of the UKESM1-0-LL stratospheric/tropospheric chemistry scheme (see Mulcahy et al., 2020, Table 2 for the list of reactions). | For the UKESM1-0-LL model, DMS concentrations are computed within the ocean biogeochemistry model MEDUSA (Yool et al., 2013) and interactively coupled with the global OGCM NEMO. DMS concentrations are linearly correlates with a composite variable that includes the logarithm of Chl a, light, and nutrients. DMS flux is calculated according to the air-to-sea gas transfer scheme of Liss and Slater (1974)[1]. DMS concentrations in the atmosphere are subsequently modified through a number of gas-phase aerosol precursor reactions within the stratospheric and tropospheric chemistry schemes of the UKESM1-0-LL model (Mulcahy et al., 2020). |

[1]The correct gas transfer scheme reference is Liss and Merlivat (1986): see Mulcahy et al., 2020, Sect. 2.4.2.

**References**

Bock, J., Michou, M., Nabat, P., Abe, M., Mulcahy, J. P., Olivié, D. J. L., Schwinger, J., Suntharalingam, P., Tjiputra, J., van Hulten, M., Watanabe, M., Yool, A., and Séférian, R. (2021). Evaluation of ocean dimethylsulfide concentration and emission in CMIP6 models. *Biogeosciences*, 18(12):3823–3860.

Brévière, E. H., Bakker, D. C., Bange, H. W., Bates, T. S., Bell, T. G., Boyd, P. W., Duce, R. A., Garçon, V., Johnson, M. T., Law, C. S., Marandino, C. A., Olsen, A., Quack, B., Quinn, P. K., Sabine, C. L., and Saltzman, E. S. (2015). Surface ocean-lower atmosphere study: Scientific synthesis and contribution to Earth system science. *Anthropocene*, 12:54–68.

Charlson, R. J., Lovelock, J. E., Andreae, M. O., and Warren, S. G. (1987). Oceanic phytoplankton, atmospheric sulphur, cloud albedo and climate. *Nature*, 326(6114):655–661. Publisher: Springer Science and Business Media LLC.

Hamilton, D. S., Lee, L. A., Pringle, K. J., Reddington, C. L., Spracklen, D. V., and Carslaw, K. S. (2014). Occurrence of pristine aerosol environments on a polluted planet. *Proceedings of the National Academy of Sciences*, 111(52):18466–18471. Publisher: Proceedings of the National Academy of Sciences.

Joge, S. D., Mansour, K., Simó, R., Galí, M., Steiner, N., Saiz-Lopez, A., and Mahajan, A. S. (2025). Climate warming increases global oceanic dimethyl sulfide emissions. *Proceedings of the National Academy of Sciences*, 122(23). Publisher: Proceedings of the National Academy of Sciences.

McNabb, B. J. and Tortell, P. D. (2022). Improved prediction of dimethyl sulfide (DMS) distributions in the northeast subarctic Pacific using machine-learning algorithms. *Biogeosciences*, 19(6):1705–1721.

Quinn, P. K. and Bates, T. S. (2011). The case against climate regulation via oceanic phytoplankton sulphur emissions. *Nature*, 480(7375):51–56.

Wang, W.-L., Song, G., Primeau, F., Saltzman, E. S., Bell, T. G., and Moore, J. K. (2020). Global ocean dimethyl sulfide climatology estimated from observations and an artificial neural network. *Biogeosciences*, 17(21):5335–5354.

Woodhouse, M. T., Carslaw, K. S., Mann, G. W., Vallina, S. M., Vogt, M., Halloran, P. R., and Boucher, O. (2010). Low sensitivity of cloud condensation nuclei to changes in the sea-air flux of dimethyl-sulphide. *Atmospheric Chemistry and Physics*, 10(16):7545–7559.

[Figure]

**Figure 1 │ Modified diagram of the climate feedback loop proposed by ref. 7.**

[Figure]

**Figure 5.** Modified diagram of the climate feedback loop of DMS.

Figure R1: Comparison of Figure 1 from QB11 (top) and Figure 5 from YW25 (bottom). The caption of Figure 1 from QB11 has been cropped: only the title is shown.

---

## Author Comment (AC1)

We sincerely thank the reviewer, Dr. Yuanxu Dong, for his thorough review of our manuscript and for the many constructive suggestions provided. Below, we respond to the comments point by point. The reviewer's comments are shown in red, our responses in blue, and excerpts from the revised manuscript are presented in *italic black*.

Air-sea gas exchange is a fundamental process in the Earth system. Dimethyl sulfide (DMS) acts as a potential climate-active gas, and its air-sea flux plays an important role in climate regulation. Understanding how surface ocean DMS concentrations and fluxes respond to future climate change is therefore essential. This study applies a novel neural network approach to investigate historical and future DMS distributions under a specific climate change scenario. Although similar studies exist, this work improves data selection for both training and projection phases and produces some distinct results. The drivers of DMS variability are further examined through sensitivity tests. The paper is well written, the topic is timely, and the method is appropriate. It could be a valuable contribution to the community after addressing the following comments.

We thank the reviewer for their positive comments and precise summary.

Line 16 – I am not a climate modeling expert, but why was SSP5-8.5 specifically chosen? Please clarify the rationale.

Thank you for your question. SSP5-8.5 represents a high-emission, fossil-fueled development pathway that induces the most pronounced changes in the climate system among the SSP scenarios. Given that oceanic DMS production and emissions are tightly coupled to marine ecosystem dynamics—which are highly sensitive to climate forcing—this scenario allows us to investigate the upper bounds of possible DMS responses. Since the primary objective of our study is to assess how DMS concentrations and fluxes might evolve under future climate change, we selected SSP5-8.5 to capture the strongest and clearest signal in the system.

Line 116 – Consider explaining the full name of CESM2-WACCM and adding a reference.

CESM-WACCM is the abbreviation of the Community Earth System Model Version 2 coupled with the Whole Atmosphere Community Climate Model. We have spelled it out in the revised manuscript (See lines 114-117), and added corresponding reference.

*"For the input data to our ANN model, we use the surface and monthly environmental outputs from CESM2-WACCM (the Community Earth System Model Version 2 coupled with the Whole Atmosphere Community Climate Model) (Danabasoglu et al., 2020)."*

Line 117 – Possibly a naive suggestion, but did you consider using the best-performing variables from different models? For example, IPSL-CM6A-LR performs well for $SiO_4$—can it be used selectively?

Thank you for this thoughtful suggestion. While certain variables—such as $SiO_4$ from IPSL-CM6A-LR—may individually perform better than those from CESM2-WACCM, we chose to use a

consistent set of input variables from a single Earth System Model (CESM2-WACCM) to maintain internal coherence and avoid potential biases introduced by mixing outputs from different models. Notably, CESM2-WACCM demonstrates the best overall agreement with observations across the key variables relevant to our study, making it a robust and internally consistent choice for driving our simulations.

Line 118 – There is no Fig. S6; this likely refers to Fig. S4.

Thank you for pointing this out. We have adjusted the order of figures and referred to the corrected figure in the text (lines 117-119).

*"This model ensemble is selected because it demonstrates the best overall results among the CMIP6 ensembles when compared to observational data (see Fig S2)."*

Table S2 – Do the last two columns represent the number of components? Please clarify.

Thank you for your suggestion. The last two columns in Table S2 represent the number of available ensemble members used to calculate the means. We have added explanations for the last two columns in Table S2.

Line 132 – A reference is needed to support this sentence.

Thank you for pointing this out. We have added a supporting reference (See lines 131-133).

*"The specific version employed, PISCESv2-gas, includes a dedicated module for simulating the cycles of climate-relevant gases (Aumont et al., 2015)."*

Line 141 – If interpreted correctly, Aranami and Tsunogai (2004) did not introduce a new parameterization but used the Nightingale et al. (2000) formulation.

Yes, indeed. We cited the wrong reference and have corrected it in the revised manuscript (See lines 140-141) as shown below.

*"DMS flux is calculated using Nightingale et al. (2000) parameterization."*

Line 142 – Please specify which gas exchange parameterization is used in this model.

Thank you for your suggestion. We have added the explanations of gas exchange parameterization in the NorESM2-LM model (See lines 145-146).

*"Sea-to-air flux DMS fluxes are calculated according to Wanninkhof (2014)."*

Line 149 – The Liss parameterization differs substantially from Wanninkhof (2014) and Nightingale et al. (2000). This likely causes significant differences in DMS flux estimation.

We partially agree with this comment. Different air–sea gas exchange parameterizations can indeed lead to notable differences in flux estimates when applied to the same DMS concentrations. To ensure consistency in comparison, we adopted the same gas transfer scheme as used in Joge et al. (2025)." (see lines 156–163).

*"As discussed above, the four ESMs employ different gas exchange parameterization schemes, which can lead to differences in estimated DMS fluxes. For instance, the scheme by Liss and Merlivat (1986) tends to underestimate fluxes at high wind speeds due to its limited representation of bubble-mediated transfer processes. In contract, the parameterization of Wanninkhof (2014) and Nightingale et al. (2000) are generally considered more reliable, as they are based on broader field datasets and better capture wind-speed dependencies. To ensure consistency in comparison, we adopted the same gas transfer scheme as used in Joge et al. (2025)."*

Line 154 – A link to Fig. S2 would be appropriate here.

Thank you for this suggestion. We have deleted the original figure S2 since it is too general and is not related to the content of the study (See lines 165-166).

*"The ANN model is a branch of artificial intelligence (AI), which builts with a fully connected network of nodes and neuron."*

Line 174 – Wang et al. (2020) is not listed in the bibliography.

Thank you for your carefully review. We have added the reference of Wang et al. (2020) in the bibliography list.

Line 211 – The value 0.6673 should be approximated as 0.67, not 0.66, for the testing/validation datasets.

Thank you. Corrected. (See lines 224-226).

*"The ANN model successfully captures the major variability in the observed data, achieving a coefficient of determination ($R^2$) of 0.68 for the training dataset and 0.67 for the testing dataset (Fig. S3)"*

Lines 224–241 – It would be informative to show the absolute concentrations and fluxes in both historical and projected periods. This would help assess whether high or low DMS regions are increasing or decreasing. Consider including these our subplots near Fig. 1 or in the Supplement.

Good points. We have followed your suggestion and added the corresponding figures in Supplement. (See Fig. S4) and also as follows,

[Figure]

**Fig. S4| Surface DMS concentrations and DMS flux over historical (1850-2014) and SSP5.85 (2015-2100) periods. a**, DMS concentrations in historical periods. **b**, DMS concentrations in SSP5.85 periods. **c**, DMS flux in historical periods. **d**, DMS flux in SSP5.85 periods.

Fig. 2 – It is notable that both this study and Joge et al. (2025) use neural networks, yet they predict systematically different historical DMS concentrations. Both presumably rely on the same observed DMS data for training. Even with different training variables, similar observed values should be reproduced, right? Please comment on this discrepancy.

Thank you for this insightful comment. We acknowledge the discrepancy and have discussed its underlying causes in our original submission. We reiterate the key points here for clarity (see lines 246-256):

*"The discrepancy arises primarily from two factors: 1) Data sources for training: Our model was trained using primarily observational variables, whereas Joge et al. (2025) used Earth system model outputs as inputs. Given the known biases in model-simulated biogeochemical fields, we consider observational data to better capture the true environmental relationships governing DMS variability. 2) Model ensemble differences: We used output from a single model (CESM2-WACCM), which demonstrates the best agreement with observations among the models considered (see Fig. S2). In contrast, Joge et al. (2025) used an ensemble of eight models, some of which show significant biases in key parameters. These differences in both training data and model ensembles likely explain the divergent historical DMS concentration patterns predicted by the two studies."*

Table 2 and Fig. 2 – Although the focus is on concentration, flux is also presented and highlighted

in the abstract. Therefore, some clarification is needed. What gas transfer velocity (K) formulation is used in Joge et al. (2025)? That study predicts lower DMS concentrations but higher fluxes. Is this due to a different K? The same issue may apply to the green and blue models in Fig. 2. Flux comparisons should use a consistent K to be comparable. Since wind speed modulates flux trends, the choice of K is likely significant. A discussion of this would be valuable near the flux results.

Following is the way that Joge et al. (2025) used to calculate the $k$ value:

$$k = \left(\frac{1}{k_w} + \frac{H}{k_a}\right)^{-1}$$

and $k_a$ is the airside transfer velocity in cm h$^{-1}$ which is calculated following Yang et al. (2013),

$$k_a = 8814 \times U^* + 6810 \times (U^*)^2$$

and U* is calculated following Johnson et al. (2010).

$$U^* = Windspeed \times \sqrt{1.3 \times 10^{-3}}$$

The waterside transfer velocity ($k_w$) in cm h$^{-1}$ is calculated following Marandino et al. (2009),

$$k_w = (0.46 \times Windspeed - 0.24) \times \frac{100}{24} \times \left(\frac{S_C}{720}\right)^{-0.5}$$

and is the Schmidt number which is calculated following Saltzman et al. (1993).

$$S_C = 2674 - 147.12 \times SST + 3.726 \times SST^2 - 0.038 \times SST^3$$

The Henry's constant (H) is the dimensionless water over air solubility of DMS and is a function of SST as follows

$$H = 0.0053 \times e^{\left(3500 \times \left(\frac{1}{SST+273.15} - \frac{1}{291.15}\right)\right)} \times 0.08206 \times (SST + 273.15) \times 101.325$$

As suggested by the reviewer, we adopt the same parameterization to calculate the flux for all models to ensure consistence.

Indeed, after adopting the same sea-to-air flux parameterization, the differences between our study and that of Joge et al. (2025) were substantially reduced. However, a noticeable discrepancy remains in the final ~20 years of the simulation: while both studies predict similar DMS concentrations during this period, Joge et al. (2025) reports higher fluxes. This difference is primarily attributable to our inclusion of sea ice coverage in the flux calculation, which was not considered in their study. Since sea ice suppresses gas exchange, especially in high-latitude regions, its omission can lead to an overestimation of DMS fluxes in those areas.

Lines 286–291 – This section requires supporting references.

Thank you for this suggestion. We have added a supporting reference in the manuscript (See lines 314-320).

*"In the VMLD experiment, DMS exhibits a decreasing trend, likely driven by the shoaling of the MLD under global warming. This shoaling MLD suppresses the upward transport of nutrient-rich deep waters (Fig. 3a), thereby limiting vertical mixing between deeper, nutrient-rich layers and the oxygenated surface ocean. As a result, primary production is reduced, particularly in low- to mid-latitude regions, ultimately leading to lower DMS concentrations at the surface ocean (Sigman &*

*Hain, 2012)"*

Section 3.3 – The first two paragraphs offer plausible explanations but would be more convincing if supported by literature. The last paragraph includes many citations, consider distributing some of these to earlier parts of the section.

Thank you for this suggestion. We have added a supporting reference in the revised manuscript.

Fig. 5 – This is a strong figure but does not reach its full potential, which is a shame. I think captions should emphasize the key message, and the main text should include dedicated discussion for this figure.

Thank you for this suggestion. We added the discussions in the updated manuscript (See lines 392-403).

*"Consistent with these findings, our results show that DMS concentrations are regionally influenced by a combination of environmental drivers (Fig. 5). Within the DMS production pathway, phytoplankton-derived dissolved DMS is negatively regulated by nutrient availability (DIN, DIP, $SiO_4$) and mixed layer depth (MLD). Its subsequent conversion to the gaseous phase and emission into the atmosphere are positively correlated with SST and surface wind speed (SSW), and negatively correlated with sea ice coverage.*
*Moreover, DMS plays a key role in the climate feedback loop. Once emitted into the atmosphere, DMS is oxidized to form non-sea-salt sulfate aerosols ($nss\text{-}SO_4^{2-}$), which act as cloud condensation nuclei (CCN). This enhances cloud albedo, reduces incoming solar radiation (positive radiative forcing), and ultimately cools the ocean surface (negative feedback)."*

Data availability – The journal likely requires a data sharing statement. Please check if the data repository link is missing.

Thank you for this suggestion. Data availability statement has been added to the text and as follows (See lines 441-445),

***"Data Availability***
*All data produced in this study are archived in the following repository: https://figshare.com/articles/dataset/Data_for_Trends_and_spatial_variation_of_oceanic_dimethy l_sulfide_under_a_warming_climate_revealed_by_an_artificial_neural_network_model/2965327 7?file=56589284."*